# Pre-grafted Group on PE Surface by DBD Plasma and Its Influence on the Oxygen Permeation with Coated SiOx

**DOI:** 10.3390/molecules24040780

**Published:** 2019-02-21

**Authors:** Hua Li, Lizhen Yang, Zhengduo Wang, Zhongwei Liu, Qiang Chen

**Affiliations:** Beijing Institute of Graphic Communication, Beijing 102600, China; BM856223@163.com (H.L.); lywzd16@163.com (Z.W.); lzwgt@126.com (Z.L.)

**Keywords:** DBD plasma, modification, PE film, barrier properties

## Abstract

In this paper, we report on polyethylene (PE) film modified by atmospheric dielectric barrier discharge (DBD) plasma prior to the deposition of SiOx coating to improve its barrier properties. Three kinds of monomers: allylamine, acrylic acid, and ethanol, are used to modify the PE surface. For comparison, Ar and O_2_ plasma pre-treatments are also performed. It is found that with the addition of a monomer in the Ar DBD plasma, the grafted active groups on PE surfaces lead to dense, pinhole-free growth of the SiOx film. The oxygen transmission rate (OTR) decreases from 700 cc/m^2^·day·atm. for the pristine to ca. 70 cc/m^2^·day·atm. for the pretreatment-coated PE, which is more than a 10-fold reduction. The relationship between the grafted monomer and the great decrease of OTR is then explored via chemical composition by attenuated total reflection Fourier transform infrared spectroscopy (ATR-FTIR) and via morphology observation by atomic force microscopy (AFM) and scanning electron microscopy (SEM). The results show that the grafted functional groups of -NH_2_, -COOH and -OH increase the surface energy and promote the nucleation of Si–O radicals on polymeric surfaces, and the formation of network and cage structures in SiOx film contributes to the significant improvement of OTR.

## 1. Introduction

Polyethylene (PE) film is one of the most popular materials used in food packaging due to its easy process, good optical transmission, and low cost [1]. However, its intrinsic poor oxygen barrier property [2] limits its usage in food packaging, especially for vacuum packaging to protect food products such as cheese, bakery goods, sliced meats, and so on. Therefore, the PE web is usually coated with nano-scale aluminum (Al) or combined with Al foil [3,4,5,6] to enhance the barrier packaging properties. As known, SiOx- coating polyethylene terephthalate (PET) demonstrates excellent properties in barrier properties, high transparency, and microwave heating process [7]. The deposition of silicon oxide on PET film as a barrier layer by electron beam evaporation, plasma vapor deposition and plasma-enhanced chemical vapor deposition (PECVD) has been widely used in the past decade [8,9,10]. Owing to the remarkably low surface energy, however, PE coated with SiOx, AlOx, or other inorganic materials have very few applications. The poor adhesion between the coating layer and substrate makes the coated PE very difficult to exhibit a high barrier property [11]. Generally, the methods to increase the PE surface energy for a hydrophilic surface are wet chemical methods [12], dry plasma treatments [13,14] and radiation treatments (ultraviolet radiation, γ ray, laser and so on) [15,16,17]. Among these approaches, plasma surface modification is one of the most appropriate and efficient approaches in terms of facile equipments, environmental friendliness, and low energy consumption [18]. Plasma generates abundant active species, including electrons, ions, reactive atoms, excited radicals, metastables, and a broad spectrum of electromagnetic radiations as well as UV light [19], and all of them can activate the plastic surface. When plasma interacts with the polymeric surface, the structure in nano-meter scale will be greatly changed while the bulk of the polymer will keep the same property. In particular, if the treatment is performed at an atmospheric pressure, e.g., atmospheric-pressure dielectric barrier discharge (DBD), which avoids using the costly traditional vacuum system, it shall be more attractive to the industry. As reported [20,21,22,23], DBD plasma can induce a moderate temperature increase while generating strongly oxidizing, long-living molecular ozone, which has significantly enabled its application in biomedical and decontamination areas. The fundamental processes of atmospheric-pressure plasma have been investigated by the transient electric field measurements and simulations of DBD in helium and the effects of conductive target contact on discharge behavior [24,25,26,27].

However, as known, the hydrophilicity of the polymer surface modified in corona discharge by using nitrogen [28], oxygen [29], or air plasmas [30], will return to the pristine status after storing for a couple of days, especially in air, which is called an “aging effect” [31]. The hydrophilicity generated by air corona discharge is temporary, and it disappears with a long storage time. Monica Pascual et al. [32] exposed the LDPE films under corona discharge before the laminated process and found that in the moderate temperatures (38 °C) and high RH (95%) condition, the T-peel strength was decreased over 9.5% percent after 15 days of aging.

Plasma modification by a monomer with the polar group instead of N_2_, O_2_, or air gas has been widely investigated and proved to be a good alternative. Gaelle Aziz et al. [33] used allylamine plasma to polymerize ultra-high molecular weight polyethylene (UHMWPE) film and kept the surface energy of UHMWPE stability for 7-day aging. Ulla Konig et al. [34] mixed H_2_O in H_2_ plasma to modify PET and got a stable 43° water contact angle (WCA) after storage in air for about 2 weeks. Annick Van Deynse et al. [35] added 2% of ethanol to the nitrogen plasma in vacuum-based DBD discharge and obtained a stable WCA at 7.5 ± 1.4° even after being stored in ambient air at a temperature of 20 °C, the relative humidity of 50% for a period of 45 days. Recently, our group also reported [36] that the treated PE was stable at a high surface energy after 3-month aging if they were treated by Ar+acrylic acid, Ar+allylamine and Ar+ethanol in an atmospheric-based roll-to-roll DBD discharge.

Following our previous work [36], in this paper we report on the deposition of SiOx coating on monomer-modified PE samples. Three monomers with different polar groups i.e., -NH_2_, -COOH, -OH, were utilized to modify the PE surface. We focus on the oxygen transmission rate (OTR) of the monomer plasma pretreatment-coated PE, not on the difference of SiOx coatings prepared in low working pressure or in atmospheric pressure. Although SiOx was frequently reported to deposit at an atmospheric pressure microplasma [37,38], specifically, as gas barrier layer on polymeric substrates by atmospheric pressure plasma jet [39,40], the deposition efficiency and little-improved oxygen barrier properties still limit the application of the atmospheric pressure plasma deposited SiOx.

## 2. Results and Discussion

### 2.1. OTR

The OTRs of the SiOx coating PE, which was pre-treated by different monomer atmospheric DBD plasmas prior to SiOx deposition, are shown in Figure 1. One can see that big differences in OTRs appeared for the untreated, Ar and O_2_ treated and monomer-treated PEs. When PE was treated by gas, i.e., Ar or O_2_ DBD plasma before coated SiOx, the OTR showed a negligible variation, which was similar to the untreated sample. While for the Ar+monomer plasma treatment, on the other hand, the OTR exhibited a significant decrease. For example, the OTR was reduced from 700 cc/m^2^·day·atm. for pristine to ca. 54.6 cc/m^2^·day·atm. for Ar + ethanol plasma treatment-coated PE. Additionally, it is obvious that the decrease of OTR depended on the monomer structure, because for Ar + ethanol, Ar+allylamine, and Ar + acrylic acid pretreatment-coated PE, the OTRs were 54.6, 74.1, and 84.6 cc/m^2^·day·atm., respectively.

SEM was used to investigate the morphology of the as-deposited SiOx. Despite the interference from the sputtered Au for good conductivity in the SEM measurement, in Figure 2, one can see that the cracks in the untreated-deposited SiOx (Figure 2a) were very obvious, while for the pretreatment-coated SiOx, on the other hand, the cracks were fine and short (Figure 2b). The reason may be that the surface modification was beneficial to the nucleation of radicals owing to the improvement of the surface energy, which could lead to the good quality of the as-deposited SiOx. It is worthy noting that with the Ar and O_2_ plasma pre-treatments, the numbers of the cracks in the as-deposited SiOx were still remarkable (images not shown in this paper).

Additionally, as reported [41,42], when polymers were treated with a carbon-rich monomer, Si–O–C or Si–C groups were easily formed in the coatings, which could provide good adhesion between the SiO_x_ layer and polymeric substrate based on the strong interfacial chemical bonding. There may be another reason for the fact that the cracks in Figure 2a, the untreatment-coated SiOx, were serious.

Untreated and treated PE surface morphologies were also examined by AFM. In Figure 3, all samples, after DBD plasma treatments, demonstrated a much rougher surface than the untreated ones, except for the O_2_ plasma treatment. The average roughness (Ra) of the pristine PE was ~30 nm. After the Ar plasma treatment, the roughness was 34.5 nm, and it was 24.5 nm after the O_2_ plasma treatment. They were 30.9 nm for Ar+allylamine, 37.2 nm for Ar+acrylic acid, and 38.8 nm for Ar+ethanol plasma treatments. The increased roughness in the Ar+monomer plasma-treated samples in Figure 3d–f were because of the combined effect of the grafted groups and plasma etching. Ar ion bombardment, a physical etching, certainly happened during pre-treatment to roughen the surface. Thus, they have a similar roughness to the pure Ar plasma-treated sample.

Regarding the relatively smooth surface in the O_2_ plasma-treated sample, it might be caused by the ashing of the polymeric in the O_2_ plasma.

Combining AFM images in Figure 3 with SEM images in Figure 2, it is affirmed that the plasma-treated surface was favorable for SiOx nucleation and growth. For the monomer-treated PEs, the Ar ion impacting renewed the PE surface in small domains, and the polymerized monomers chemically interacted with the PE surface and bridged the nucleated Si–O species to form a strong bond at the interface of the PE and SiOx layer. As a result, the SiOx layer was adhered solidly on the surface and hardly any cracks formed. For untreated PE, on the other hand, the low surface energy surface physically adsorbed the diffusion radicles from plasma, which had nucleated and grown for SiOx film in irregular. Thus, the adhesion between SiOx and the PE surface was definitely poor, and the cracks were formed in coated SiOx when they were grown continuously.

### 2.2. WCA

The improvement of hydrophilicity (surface energy) after plasma treatment was examined by WCA measurements. As known, the hydrophilic surface is essential for the dense nucleation of radicals. In Figure 4, one can see that PE hydrophilicities after plasma treatment, i.e., Ar, O_2_ and Ar+monomers were improved. The WCAs were decreased from 102.4° for pristine PE to 57.1°, 55.9°, 51.4°, 53.6°, and 60.2° for Ar, O_2_, Ar+allylamine, Ar+acrylic acid, and Ar+ethanol plasmas, respectively, i.e., the surface was switched from hydrophobic to hydrophilic. However, as previously reported [35], with the monomer’s addition to the argon carrier gas on a 95% confidence interval, the monomer’s addition had no significant influence on the WCA value.

### 2.3. ATR-FTIR Analysis

With ATR-FTIR, we examined the PE surface to explore if the grafted active groups on the PE played a vital role in Si­­–O species nucleation and led to the improvement of OTR. In Figure 5, one can see that all samples show peaks from the PE bulk structure, the peaks at 600 cm^−1^–1160 cm^−1^ for the PE skeleton structure, -C-C- stretching vibration at 2915 cm^−1^, and the peaks at 2847 cm^−1^, 1462 cm^−1^, and 719 cm^−1^ for -CH_2_ asymmetric stretching vibration, symmetric stretching vibration, deformation vibration and rocking vibration [43], respectively. It is confirmed that the PE bulk property was not damaged after DBD plasma treatment. After the monomers, i.e., ethanol and acrylic acid, were used to treat PE, as Figure 5 shows, the new peaks at 3415 cm^−1^ and 1620 cm^−1^, which were assigned to -OH stretching and bending vibration [44] and the obvious peak at 1720 cm^−1^ for C=O stretching vibration in –COOH [44], were observed. For the allylamine modified sample, new peaks for the -NH stretching vibration were visible around 3367 cm^−1^, 3295 cm^−1^, 1633 cm^−1^, and C-N at 1337 cm^−1^ [45]. The new obvious peaks after monomer treatment in the FTIR spectrum confirmed the incorporation of functional groups -OH, -COOH, and -NH_2_ from corresponding monomers on the PE surface. This supports our previous assumption that the polar groups on the surface were mainly responsible for the good hydrophilicity in Figure 4.

The SiOx chemical component also dominates its properties; the relationship between different surface groups with the SiOx structure was then analyzed. Figure 6a shows that except for the absorption peaks of PE at 2915 cm^−1^, 2847 cm^−1^, 1462 cm^−1^, and 719 cm^−1^ for -C-C- stretching vibration in each sample, the obvious new peaks appearing at 1065 cm^−1^ and 810 cm^−1^ correspond to the asymmetric stretching vibration of the Si–O–Si bond [46], which caused the formation of SiOx on all PE surfaces. The main peak of Si–O–Si was at 1065 cm^−1^ rather than at 1070 cm^−1^ [7], which means that the SiOx coating was impure and the C and H from the monomer were incorporated into the coating.

However, for the untreated PE substrate, the peaks at around 3300 cm^−1^ and 930 cm^−1^ were observed in the spectrum, which correlated to the SiOH stretching and bending vibrations, respectively. It was assumed that they were caused from the inactive surface physical adsorption of the species in the plasma. The SiOH concentration was increased along with exposure time due to the absorption of water vapor from the ambient, which would increase the void volume in the as-deposited SiOx and lead to the short path of the gas permeation crossing the coating [47].

For Ar and O_2_ modifying samples, Figure 6a shows that not only the ratio of 1065 cm^−1^ and 810 cm^−1^ for the Si-based peak was smaller, but the shoulder peak at 1240 cm^−1^ was shifted to a high wavenumber. It means that the as-deposited SiOx coating contained a high concentration of the organic component. The symmetric bending vibration mode of the Si–CH_3_ bond at 1277 cm^−1^ indicated the incomplete oxidation reaction of O_2_ plasma with HMDSO and the presence of hydrocarbon radicals in the coatings. The high concentration of the organic component in SiOx coatings definitely raised the OTR value.

Regarding all Ar+monomer modifying samples, one can notice that in Figure 6a all peaks of the Si-based groups were remarkable and show obvious variation. The shoulder moved from 1277 cm^−1^ to 1200 cm^−1^, and the ratios of peaks at 1065 cm^−1^ and 810 cm^−1^ were dependent on the grafted surface, and the peak at 960 cm^−1^ for Si–CH_3_ was remarkable and different in three samples. Therefore, we concluded that the different surface groups certainly induced the alternation of the as-deposited SiOx chemical component and structure.

SiOx has three chain structures, and a different structure demonstrated a different efficiency on the barrier properties [48,49,50]. We then deconvoluted the Si–O–Si asymmetric stretching peak in the range of 950~1250 cm^−1^ into three peaks at 1023 cm^−1^, 1063 cm^−1^ and 1135 cm^−1^ for linear, network and cage structures, respectively [51], as shown in Figure 6. The peak centered at 1135 cm^−1^ is attributed to larger angle Si–O–Si bonds in a cage structure with a bond angle of approximately 150°. The peak centered at 1063 cm^−1^ is attributed to the stretching of a smaller angle Si–O–Si bond in a network structure. The peak at 1023 cm^−1^ is attributed to the stretching of an even smaller Si–O–Si bond angle, encountered in a linear silicon suboxide. The calculated three peak areas derived from Figure 6b–g are listed in Table 1.

As seen, if SiOx was deposited on Ar and O_2_ pretreatment samples, the films contained a high ratio of linear structure, while for the monomer-grafted samples, on the other hand, the linear structure was very small, even the linear structures disappeared in the spectrum. In contrast, the cage and network structures of Si–O–Si were dominant in three monomer-modified samples.

Combined with Figure 1, it can be concluded that the highest proportion of network structure in ethanol pre-treatment-coated SiOx might be responsible for the minimum OTR in ethanol pre-treatment-coated PE. The growth of the network and cage Si–O–Si structures replacing the linear structure in the SiOx chemical structure was more advantageous in gas permeation resistance than that by the linear structure.

In Table 1, one can also notice that the as-deposited SiOx on the untreated PE surface also had a relatively high proportion of cage and network, but the linear structure of Si–O–Si was formed, and the proportion of network and cage structures was relatively small in the whole component. It might be one of the reasons that the untreated SiOx-coated PE exhibited a high OTR.

As aforementioned, the SiOH group in the untreated sample increased the void volume in the as-deposited SiOx, which led to a short path of gas permeation crossing the coating and increased OTR value.

Additionally, the poor adhesion between the as-deposited SiOx and untreated PE, as unveiled in the Figure 2 SEM image, was also responsible for the poor barrier property to oxygen gas.

Thererfore, we now can conclude that a different surface group can induce a different structure of Si–O–Si growth of SiOx film on a PE surface.

In Figure 6a, the weak peak at 910 cm^−1^ assigned to the Si–N [52] in allylamine-modified PE was considered from the incorporated nitrogen, which can release the inner residual stress in the as-deposited SiOx [53] and be beneficial to a small OTR as shown in Figure 1.

## 3. Materials and Methods

### 3.1. Experiment Setup

For the polymeric surface modification, an atmospheric pressure roll-to-roll DBD setup was utilized, which has been described in other papers [36,54]. Here we give a brief introduction. The roll-to-roll DBD setup consisted of a power supply, roll-to-roll system, dielectric covered stainless-steel electrodes with a water cooling system, gas delivery system, monomer feeding system, and off-gas exhaust system. A 20 kHz, 500 W sinusoidal power supply was applied to cooled DBD electrodes, where each electrode was covered by 2 mm in thickness, 290 × 80 mm^2^ in the area quartz plane as dielectric. The distance between two electrodes was fixed at 2 mm. The monomer was inlet into the space by carrier gas Ar, where Ar, one role, was used as the carrier gas to control the monomer amount in the space by flowing rate meters; the other role was as discharge gas to improve the discharge uniformity in glow or quasi-glow discharge mode.

For SiOx deposition, on the other hand, the process was performed at a low working pressure. The setup was a capacitively coupled plasma (CCP), where the plasma was generated between electrodes, as shown in Figure 7. A 40 kHz frequency power supply was used to generate plasma in pulse mode with a duty cycle of 20% during the PECVD process.

### 3.2. Treatment Processes and Materials

Prior to the generation of plasma for treatment, the discharge gas Ar and monomer, which was also carried by Ar, were mixed and inlet the space between electrodes for 3 min. The parameters of DBD treatments are listed in Table 2. The flow rates of discharge gas and carrier gas were fixed at 400 L/h and 30 L/h, respectively. During plasma treatment in the roll-to-roll process, the web speed was set at 10 m/min. Allylamine, acrylic acid, and ethanol, three monomers, were of analytically pure grade. The discharge gas and carrier gas Ar were 99.999% in purity (Beijing Praxair Ltd., Beijing, China). The PE web, 45 μm in thickness and 200 mm in width, was a commercial product without any pre-treatment.

After DBD modification, the samples were cut into ca. 20 × 20 cm^2^ sheets and then deposited on a 150-nm-thick SiOx coating by CCP in batch mode, where the ratio of HMDSO and O_2_ was set at 1:1, the working pressure was 15 Pa, the applied power was 30W, and the power duty cycle was 20%. Note that only the DBD modified side of the PE web was deposited with SiOx. Because the monomer modified sample can remain at the high surface energy for over 3 months [36], we did not intentionally fix the period between the plasma treatment and the SiOx deposition. Normally the treated samples were put into the chamber for PECVD SiOx after 2–3 h.

### 3.3. Characteristics of Samples

The hydrophilicity of the modified PE films was examined by water contact angle (WCA) measurement. The WCA values were the average of five samples, and the error was in the range of ±2°. The surface components were analyzed by attenuated total reflection Fourier transform infrared spectroscopy (ATR-FTIR) with a KRS-5ART crystal after 64 scanning repetitions. For morphology, SEM (BCPCAS4800, JEOL, Tokyo, Japan) and AFM (SPA300, Veeco, NY, USA, tapping model with 256 pixels resolution) were utilized. The OTR of SiOx coating-modified PE was measured in Oxygen Permeation Analysers(8001, Systech Illinois, Johnsburgy, IL, USA) with three averaged samples, and the deviation of the OTR value was controlled at ±5%.

## 4. Conclusions

In this paper, we achieved a small O_2_ permeation rate in DBD plasma treatment-coated SiOx PE, from 700 cc/m^2^·day·atm. for pristine to ca. 70 cc/m^2^·day·atm. for monomer-modified PE. With ATR-FTIR analysis, we found that monomer DBD plasma-treated PE could obviously decrease the SiOH formation in SiOx film. In addition, the active groups from the grafted monomer could suppress the growth of the linear structure of Si–O–Si and induce the formation of network and cage structure of Si–O–Si in SiOx films. It is one of the main reasons that the SiOx coating demonstrated high barrier properties. Another contribution of the monomer plasma-grafted PE to improve the high OTR of PE was the high surface energy after plasma treatment, which benefited the nucleation of radicals and the high adhesion of the coating to the PE substrate. It was confirmed by AFM and SEM images that the short and fine cracks were formed on the pretreatment-coated SiOx film. We can then conclude that the monomer modification of PE promoted the nucleation and adsorption of Si–O precursors owing to the high surface energy, and induced compact SiOx growing in the network and cage structure, contributing to the small oxygen permeation rate.

## Figures and Tables

**Figure 1 molecules-24-00780-f001:**
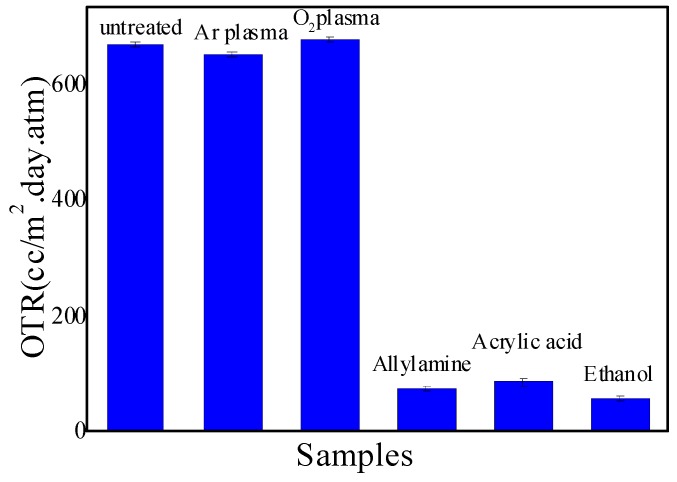
The dependence of OTRs of SiOx coating PE on the monomers.

**Figure 2 molecules-24-00780-f002:**
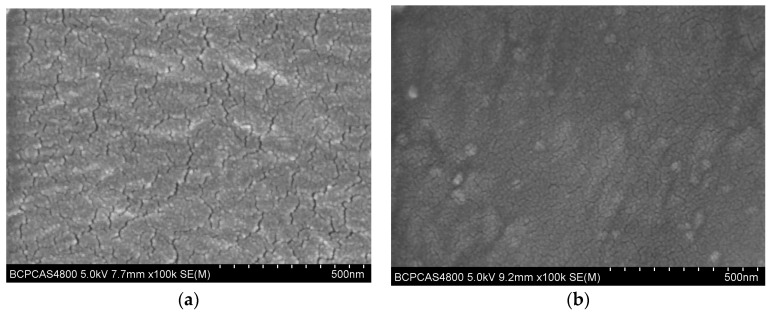
SEM morphology of SiOx coatings grown on **a**- preistine PE, **b**- Ar+ ethanol plasma-treated PE+ SiOx.

**Figure 3 molecules-24-00780-f003:**
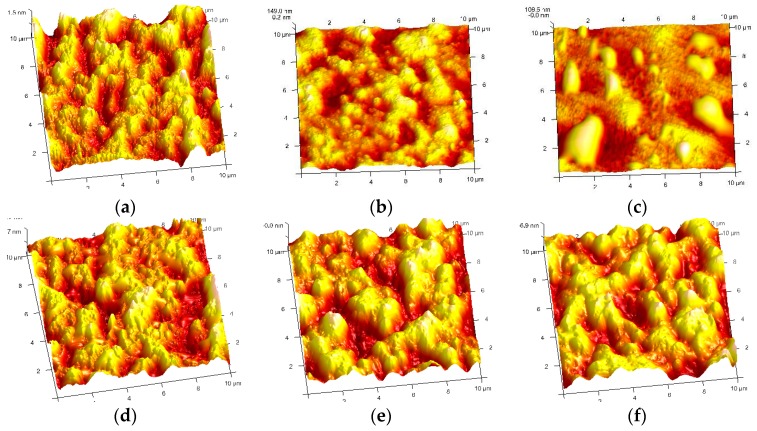
AFM image of PE films after different DBD plasma treatments (**a**: pristine PE, Ra 30nm; **b**: Ar plasma, Ra 34.5 nm; **c**: O_2_ plasma, Ra 24.5 nm; **d**: Ar+allylamine plasma, Ra 30.9 nm; **e**: Ar+acrylic acid, Ra 37.2 nm; and **f**: Ar+ethanol, Ra 38.8 nm).

**Figure 4 molecules-24-00780-f004:**
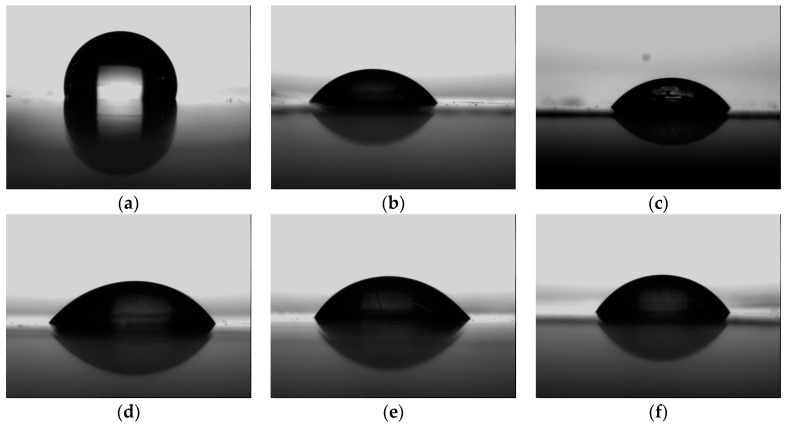
WCAs of DBD pre-treated PEs (**a**: pristine PE, 102.4°; **b**: Ar plasma, 57.1°; **c**: O_2_ plasma, 55.9°; **d**: Ar+allylamine, 51.4°; **e**: Ar+acrylic acid, 53.6°; **f**: Ar+ethanol, 60.2°).

**Figure 5 molecules-24-00780-f005:**
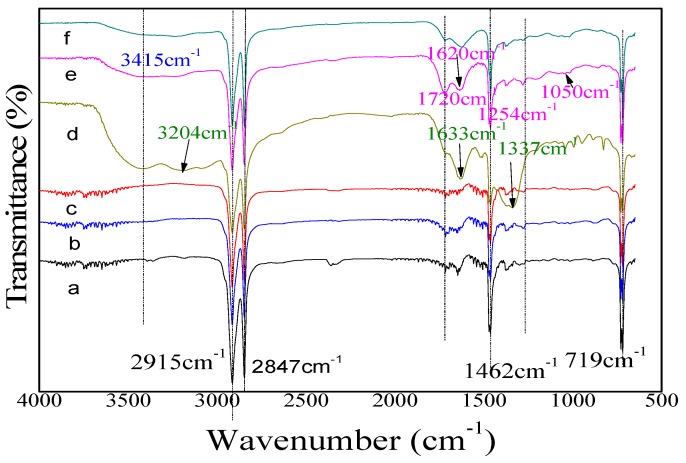
ATR-FTIR spectroscopy of PE (**a**: pristine PE) and PES after DBD pre-treatments (**b**: Ar plasma; **c**: O_2_ plasma; **d**: Ar+allylamine; **e**: Ar+acrylic acid; **f**: Ar+ethanol).

**Figure 6 molecules-24-00780-f006:**
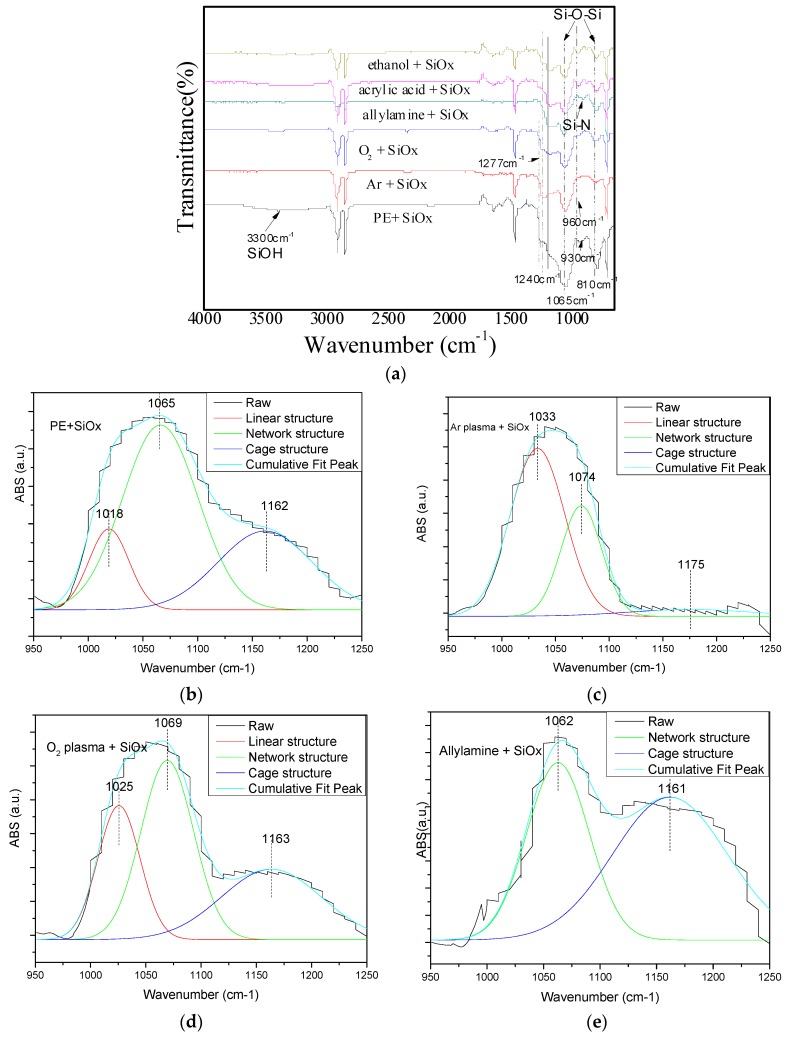
ATR-FTIR spectroscopy of SiOx coating with different DBD pre-treatment PE sheets as substrates (**a**: the survey of SiOx coatings deposited on different pre-treated PEs. The deconvolution of Si–O in the range of 950 cm^−1^–1250 cm^−1^ in **b**- PE + SiOx, **c**-Ar plasma + SiOx, **d**- O_2_ plasma + SiOx, **e**- Ar + llylamine + SiOx, **f**- Ar + acrylic acid + SiOx, and **g**-Ar + ethanol + SiOx.).

**Figure 7 molecules-24-00780-f007:**
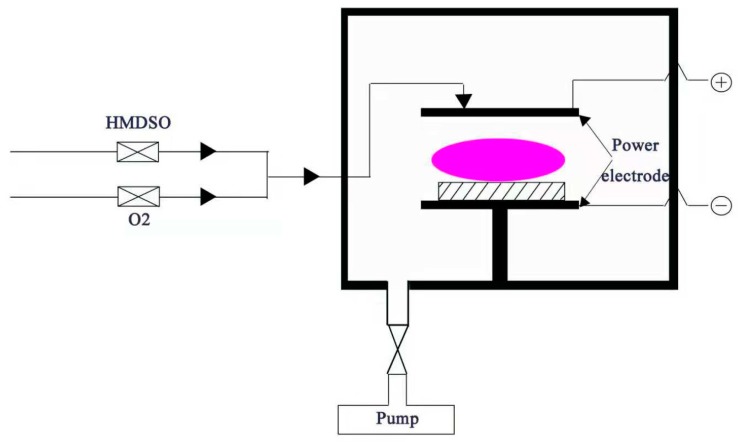
The schematic of the PECVD equipment.

**Table 1 molecules-24-00780-t001:** The ratio of deconvoluted Si–O–Si in three peaks at 1023 cm^−1^, 1063 cm^−1^ and 1135 cm^−1^ for linear, network and cage micro-structures, respectively.

Samples	Linear Structure	Network Structure	Cage Structure
PE+ SiOx	0.132	0.567	0.303
Ar pretreatment + SiOx	0.632	0.310	0.058
O_2_ pretreatment + SiOx	0.255	0.427	0.318
Ar + allylamine pretreatment + SiOx	0	0.404	0.596
Ar + acrylic acid pretreatment + SiOx	0	0.502	0.498
Ar + ethanol pretreatment + SiOx	0	0.596	0.404

**Table 2 molecules-24-00780-t002:** The process parameters of DBD plasma treatment.

Sample	Gas/Monomer	Discharge Ar (L/h)	Carrier Ar (L/h)	PE Web Speed (m/min)
1	Ar	400	-	10
2	O_2_	400	-	10
3	Ar + allylamine	400	30	10
4	Ar + acrylic acid	400	30	10
5	Ar + ethanol	400	30	10

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
