# Peer review of "Pre-grafted Group on PE Surface by DBD Plasma and Its Influence on the Oxygen Permeation with Coated SiOx"

_molecules, 2019, doi:10.3390/molecules24040780_

Reviewer 1 Report

The authors are suggested to get editing help from someone with good written English skill in order to improve the readability of this manuscript. For example, in the Title (line 3), "...With Coated SiOx" is better than "...After Coated SiOx". In line 13, it should be "... with monomer addition..." instead of "... with monomer added...".  In line 14, it should be "... growth of the SiOx film" instead of "... growth of the SiOx film growth".  In line 14, it should be "... oxygen transmission rate (OTR)" instead of "... oxygen transmission rates (OTRs)". In line 15, it should be "decreases..." instead of "decease...". In line 15, it should be "... for the pristine...for the pretreatment-coated..." instead of "... for pristine...for pretreatment-coated...". In line 16, it should be "... relationship between...and..." instead of "... relationship of...with...". In line 17, it should be "... explored via chemical analysis by..." instead of "... explored with the chemical composition through...". In line 18, it should be "... and via morphology observation by..." instead of "... and morphology by...".

Moreover, there are other more important issues as listed below:

1. The "aging effect" mentioned in line 51 is not addressed at all in this study.

2. The choice of the process parameters in Table 1 is not justified. A systematic study of the effects of the process parameters is missing.

3. The argument regarding Fig. 3 is not convincing. The comparison of the cracks is kind of arbitrary and far from exact. The authors are suggested to quantify the data for comparison.

4. Similarly, the argument regarding Fig. 4 is not convincing. How can one say a Ra of 30.9 nm is "much rougher" than that of 30 nm?

5. Similarly, the argument regarding Fig. 5 is confusing. The dipole moment of -NH2, -COOH, and -OH is 1.46 D, 1.63 D, and 1.7 D respectively. How can the authors say -NH2 is the one with the highest polar strength?

6. The results described in Sec. 3.3 require further double check on both their accuracy and correspondence with the data shown in the Figures. The subsequent and lengthy discussion needs to be carefully organized and soundly argued based on clearly referred supporting literature. Otherwise, it can become very confusing and lack of convincing.

7. The issue of adhesion between the SiOx layer and polymeric substrate argued in line 142 can be easily clarified by scratch test, rather than proof-less hypothesis.

Author Response

Dear Reviewer,

Thanks very much for your reply. Here we have added attachments of our revised article, which has been revised according reviewers’ comments with highlight of revision and the other file with name “answer to reviewers”, where point-by-point replies to the reviewers’ comments and either a list of changes we have made is highlighted by using colored text. Hopefully, you can review it again and publish it soon in your good reputation journey, Molecules.

Thanks!

Sincerely your

Hua Li

Reviewer 2 Report

Pleaserefer to attached file with 7 comments. Special care should be addressed to C1, C3, C6 and C7.

Author Response

Dear Reviewer,

Thanks very much for your reply. Here we have added attachments  with name “answer to reviewer 2”, where point-by-point replies to the your comments and either a list of changes we have made is highlighted by using colored text. Hopefully, you can review it again.

Thanks!

Sincerely your

Hua Li

Round  2

Reviewer 1 Report

First of all, the authors did not take my suggestion "to get editing help from someone with good written English skill in order to improve the readability of this manuscript". Yielding merely to what I recommended in the abstract, which I used as an EXAMPLE only, does not equalize to a "complete revision" in English. Even the reply from the authors as well as the revised parts (in red in the revised manuscript) are full of inadequate use of English.

Secondly, most, if not all, of my serious opinions about the original manuscript are "lightly deviated/ignored" in the revised one. For example, most of the SEM data shown in Fig. 3 disappear in the revised version. The arguments about Figs. 4 and 5 are largely shortened and omitted to avoid serious discussion. Not to mention the rest.

Overall, the authors are urged to face their own "complete" experimental results, not to hide them, and seriously revise the manuscript. I will not recommend its acceptance otherwise.

Author Response

Dear Reviewer,

Thanks very much for your reply. Here we have added attachments of our second revised article, which has been seriously revised according your comments with highlight of revision and the other file with name “answer to reviewer”, where point-by-point replies to your comments and either a list of changes we have made is highlighted by using colored text. Hopefully, you can review it again and give us your suggestion, thanks very much for your attention and hope you can publish it soon in your good reputation journey, Molecules.

Thanks!

Sincerely your

Hua Li
